# Identification of the immune-associated enhancer RNA *SATB1-AS1* as a novel biomarker for thymic cancer prognosis

**Hongpeng Wang**[1⊙], **Yichu Huang**[2⊙], **Guangtao Min**[1], **Lei Jiang**[1]*

**1** Department of General Surgery, The First Hospital of Lanzhou University, Lanzhou, Gansu, P.R. China,
**2** The First School of Clinical Medicine, Lanzhou University, Lanzhou, Gansu, P.R. China

⊙ These authors contributed equally to this work.
* jiangyjsggyx@163.com

## Abstract

### Objective

To screen key enhancer RNAs (eRNAs) in thymoma (THYM) through The Cancer Genome Atlas (TCGA) database and explore their potential as prognostic molecules and therapeutic targets in THYM.

### Methods

Gene expression RNA-seq profiles of 33 cancer types were retrieved and downloaded from the TCGA database, and Kaplan–Meier survival analysis and Spearman correlation analysis were applied to screen eRNAs and target genes associated with survival in THYM patients. The correlation of target eRNAs with clinical features was assessed. Gene set enrichment analysis was performed to investigate the potential biological functions of target eRNAs. Finally, the prognostic potential of the eRNA was validated in other cancer types.

### Results

In THYM, *SATB1-AS1* and the target gene SATB1 high expression were associated with a positive patient prognosis, and *SATB1-AS1* expression was negatively correlated with patient stage ($P < 0.05$). Gene and pathway enrichment analyses revealed that *SATB1-AS1* was associated with leukocyte transendothelial migration, natural killer cell-mediated cytotoxicity, neutrophil extracellular trap formation, PD-L1 expression and the PD-1 checkpoint pathway in cancer.

### Conclusion

*SATB1-AS1* may be a key eRNA in THYM and has the potential to be a marker and therapeutic target for the early diagnosis and prognosis of THYM.

**Data availability statement:** The RNA expression profiles and corresponding clinical information used in this study are available in the TCGA (https://portal.gdc.cancer.gov) databases. We first screened 124 THYM-related patients and 14 normal sample via the TCGA website and downloaded their transcript data (Project Number: TCGA-THYM). We also used the UCSC Xena online database (https://xena.ucsc.edu/) to obtain clinical data and expression profiling data for 33 cancers. All relevant data are within the paper and its Supporting Information files.

**Funding:** This work was supported by the National Natural Science Foundation of China (grant no. 82060527), Science Foundation of the First Hospital of Lanzhou University (no. ldyyyn2019-02), Medical Research Improvement Project of Lanzhou University (no. lzuyxcx-2022-175) awarded to LJ.

**Competing interests:** The authors have declared that no competing interests exist.

# 1 Introduction

Thymoma (THYM) is the most common tumor type in the anterior mediastinum, accounting for approximately 95% of all thymic tumors, with an incidence rate of 4.09 per million in China, which is much higher than that in other countries in the world, including Europe [1]. Milan Radovich et al. systematically classified thymomas and reported that histologic subtypes A, AB, B1, B2, B3, and thymic carcinoma (TC) were significantly associated with multiple gene mutations [2]. Notably, among many advanced tumors, the 5-year median survival rate for thymomas is 69%, whereas the 5-year median survival rate for TC is only 36% [3]. Surgery, as the treatment of choice for tumor therapy, is the only choice for patients with early-stage thymoma, whereas palliative treatments such as radiotherapy have become the only option for patients with advanced unresectable tumors [4]. In recent years, the emergence of immunotherapy and targeted therapy has provided many new insights into the treatment of patients with thymomas and has also shown some therapeutic promise [5]; however, further development of more individualized treatment regimens to increase disease responsiveness is still needed.

Enhancers are DNA elements that bind to cofactors and transcription factors (TFs), are located in noncoding regions of the genome, have a characteristic chromatin structure, and are involved in driving the transcription of target genes in a variety of ways, such as three-dimensional chromatin remodeling and enhancer–promoter loops [6]. With the development and advancement of genome sequencing and large-scale genome-wide association studies (GWASs), it has been demonstrated that active enhancers can transcribe noncoding RNAs (ncRNAs) called enhancer RNAs (eRNAs) and that eRNA-mediated enhancer activity plays a definitive role in the progression of many cancers [1]. It has been shown that long non-coding RNA (lncRNAs) (*NR2F1-AS1, LINC00665, and RP11-285A1.1*) and microRNAs (hsa-miRNA-143, hsa-miRNA-141, hsa-miRNA-140, and hsa-miRNA-3199) are significantly correlated with prognosis and overall survival in THYM [7]. Genetic variation in eRNA transcripts was analyzed in more than 30 different types of cancers from TCGA via eRNA quantitative trait loci (eRNAQTLs), confirming that the activation of enhancers and the transcription of eRNAs usually induce mutations in oncogenes or the activation of oncogenic signaling pathways [8]. A recent study revealed that eRNA (CRISPR-Cas9) can control the oncogenic activity of the MYB and DCTD genes and is a key determinant of B-cell precursor acute lymphoblastic leukemia (B-ALL) [9].

However, studies on eRNAs are still limited, and the functions and roles of most eRNAs are still unclear. THYM-associated eRNA regulatory mechanisms are largely worth exploring. Here, we aimed to screen prognostic eRNAs and their target genes in THYM. We found that the eRNA *SATB1-AS1* was significantly associated with the survival of THYM patients through regulating SATB1 and was correlated with the local immune environment of THYM, which is expected to be a new biomarker for THYM prognosis and immunotherapy response.

## 2 Materials and methods

### 2.1 Acquisition of the THYM of eRNA data

The Cancer Genome Atlas (TCGA) provides most of the expression data of cancer patients and their clinical data (https://portal.gdc.cancer.gov/) [10]. We first screened 124 THYM-related patients and 14 normal sample via the TCGA website and downloaded their transcript data (Project Number: TCGA-THYM). Next, the transcript IDs were converted to gene symbol IDs via the pl language and differentiated between tumor patients and controls. We also used the UCSC Xena online database (https://xena.ucsc.edu/) to obtain clinical data and expression profiling data for 33 cancers. The data extraction deadline is August 24, 2023. The study was conducted in accordance with the Declaration of Helsinki (revised 2013).

### 2.2 Screening for THYM-related differential eRNAs

The R package "limma" [11] was used to screen for differentially expressed eRNAs, and patients were categorized into low- and high-expression groups on the basis of the median expression value of each eRNA. The R packages "survival" and "survminer" were used to construct a loop function to perform a log-rank test for each eRNA and plot the Kaplan–Meier survival curve [12]. $P < 0.05$ was considered statistically significant.

### 2.3 Identification of THYM coexpressed genes and gene set enrichment analysis of prognostically relevant eRNAs

Genes with correlation coefficients $R > 0.7$ and $P < 0.001$ were considered coexpressed genes of prognostically associated eRNAs. Functional enrichment analysis of the coexpressed genes revealed their biological functions in THYM. Gene Ontology (GO) and Kyoto Encyclopedia of Genes and Genomes (KEGG) enrichment analyses of coexpressed genes for prognosis-associated eRNAs were performed via the R software package "clusterProfiler" and then visualized via the online analysis website bioinformatics.

### 2.4 Analysis of common mutations of SATB1 in THYM

The TCGA database provides exome sequencing data of nearly 30 tumors, and we classified 119 samples with mutations into two groups according to the expression of SATB1, the target gene of *SATB1-AS1*, and compared the mutation frequency of related genes between the two groups to calculate the tumor mutation burden (TMB) [13]. The R package "maftools" was used for cluster analysis, and finally, the R package "waterfall" was used to draw waterfall plots to visualize the data.

### 2.5 Correlation analysis of *SATB1-AS1* immune checkpoints and immunoregulatory genes

For the correlation analysis of immune-related genes, we extracted the ENSG00000182568 (SATB1) gene and 60 genes of two types of immune checkpoint pathways (inhibitory (24), stimulatory (36)) and 150 genes of five types of immune pathways (chemokine (41), receptor (18), MHC (21), immunoinhibitor (24), and immunostimulator (46)) [14]. The results were filtered out by screening all normal samples, and a log2 (x + 0.001) transformation was applied to each expression value, which ultimately resulted in a sparse correlation between SATB1 and immunomodulatory genes in a variety of cancers.

### 2.6 *SATB1-AS1* pancancer survival and correlation analysis

Patients with 32 other cancers were categorized into low and high *SATB1-AS1* expression groups according to their median *SATB1-AS1* expression values via the R package "limma". The prognostic value of *SATB1-AS1* was investigated

via the Kaplan–Meier method, and the correlation between *SATB1-AS1* and pancancer was evaluated via the Spearman correlation test.

## 2.7 Statistical analysis

The data were statistically analyzed via R software (version 4.4.1). Spearman correlation was applied to estimate the strength of correlation between two samples. The Shapiro-Wilk test is used to test the normality of the data distribution. Survival analysis was performed via the Kaplan–Meier method, and comparisons between clinical variables in the two groups were performed via the Wilcoxon rank sum test. $P < 0.05$ was considered statistically significant.

# 3 Results

## 3.1 Screening for eRNAs associated with THYM survival

Patient eRNA expression data and clinical survival-related information were combined, and the data were normalized (log2 (x + 0.001)), while information on patients with eRNA expression of 0 and normal controls was removed. With eRNA expression divided into high and low groups according to the median expression value, 133 eRNAs significantly associated with OS were identified via the Kaplan–Meier log-rank test ($P < 0.05$) (S1 Table). After further screening, only 10 genes were significantly correlated with the predicted mRNA levels of the target genes (coefficient of r > 0.7, $P < 0.001$), as shown in Fig 1.

## 3.2 The eRNA *SATB1-AS1* had the highest positive correlation with its target gene SATB1 in THYM prognosis

Among the 10 prognostically relevant eRNAs, *SATB1- AS1* had the highest prognostic correlation with THYM (Fig 2A). Therefore, *SATB1-AS1* was identified from the 10 eRNAs in this study. We found that high *SATB1-AS1* expression was associated with a favorable prognosis in THYM patients. Furthermore, lower expression of SATB1-AS1 is more associated with tumor stage and lymphatic invasion compared to higher expression (Fig 2B). Notably, the expression level of

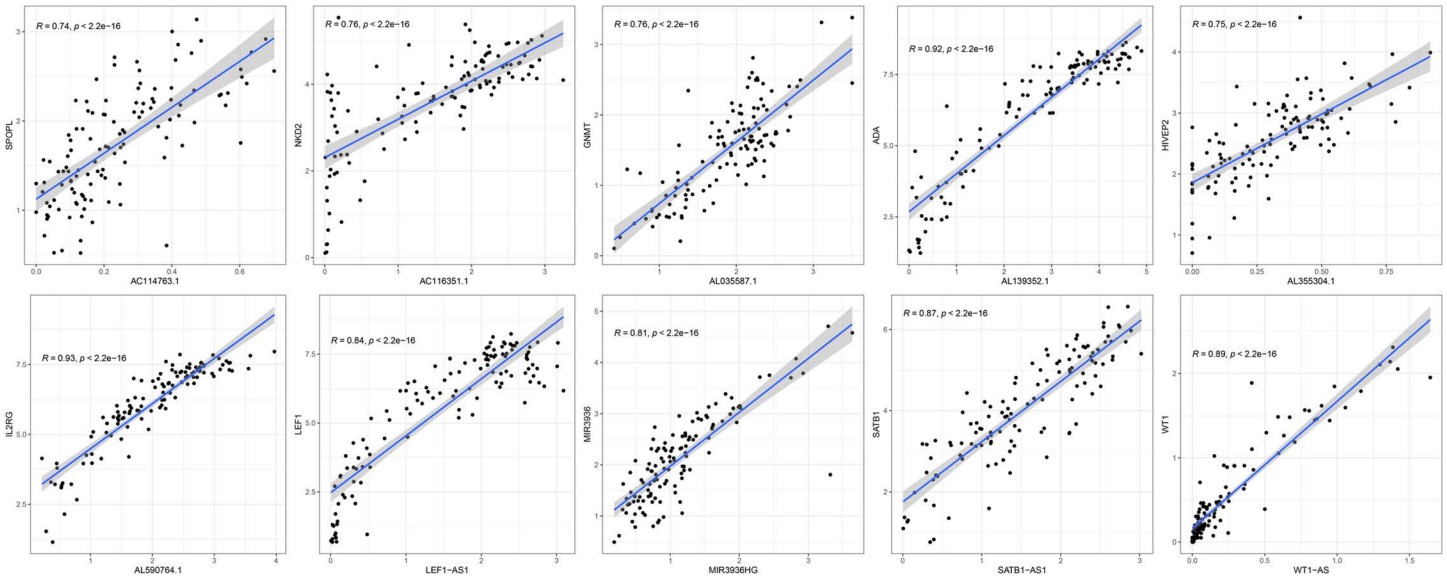

**Fig 1. Scatterplot of correlation analysis showing 10 eRNAs strongly associated with THYM survival.**

SATB1-AS1 is unrelated to age, gender, and cancer status (Fig 2C-E). These results suggest that *SATB1-AS1* may be a favorable independent prognostic biomarker for THYM patients.

### 3.3 *SATB1-AS1* is associated with multiple immune-related signaling pathways

GO enrichment revealed that in the Biological Process (BP) category, *SATB1-AS1* was associated with chromosome segregation, nuclear division and organelle fission (Fig 3A and B). In Cellular Component (CC), *SATB1-AS1* is involved mainly in the T-cell receptor complex, plasme membrane signaling receptor complex and chromosomal region. In Molecular Function (MF), *SATB1-AS1* was involved mainly in microtubule binding, protein serine/threonine kinase activity and tubulin binding. The KEGG signaling pathway was also enriched in several signaling pathways related to immune cells, e.g., leukocyte transendothelial migration, natural killer cell-mediated cytotoxicity, neutrophil extracellular trap formation, PD-L1 expression and the PD-1 checkpoint pathway in cancer, the T-cell receptor signaling pathway, Th1 and Th2 cell differentiation and Th17 cell differentiation. Thus, we performed a Spearman correlation analysis between SATB1, the target gene of *SATB1-AS1*, and immune-related genes and found that the expression of SATB1 was significantly associated with several immune checkpoint marker genes as well as immune immunization pathway-related genes (Fig 4A and B).

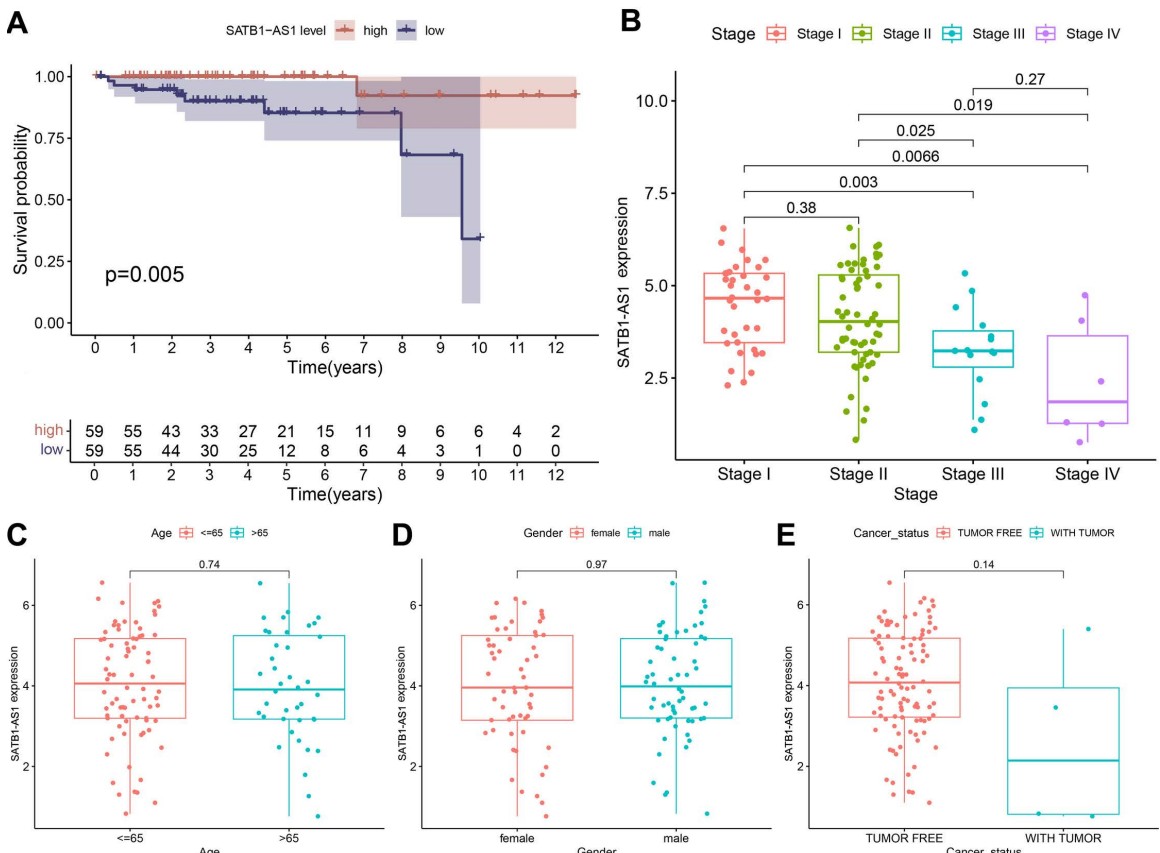

**Fig 2. The clinical relevance of eRNA *SATB1-AS1* to THYM.** (A) Kaplan–Meier OS curve for THYM patients with *SATB1-AS1*-high and *SATB1-AS1*-low expression. (B-E) Clinical relevance of eRNA *SATB1-AS1* expression in THYM patients.

## 3.4 The mutational landscape of SATB1

TMB could serve as an emerging tumor immunotherapy biomarker [15], so we assessed the differences in the frequency of mutations in each set of samples via chi-square tests. In THYM, the most relevant and frequent missense mutations associated with SATB1 were those in GTF2I and HRAS (Fig 5A). Fig 5B shows the relationship between SATB1 mRNA expression and its putative copy number alterations (CNAs).

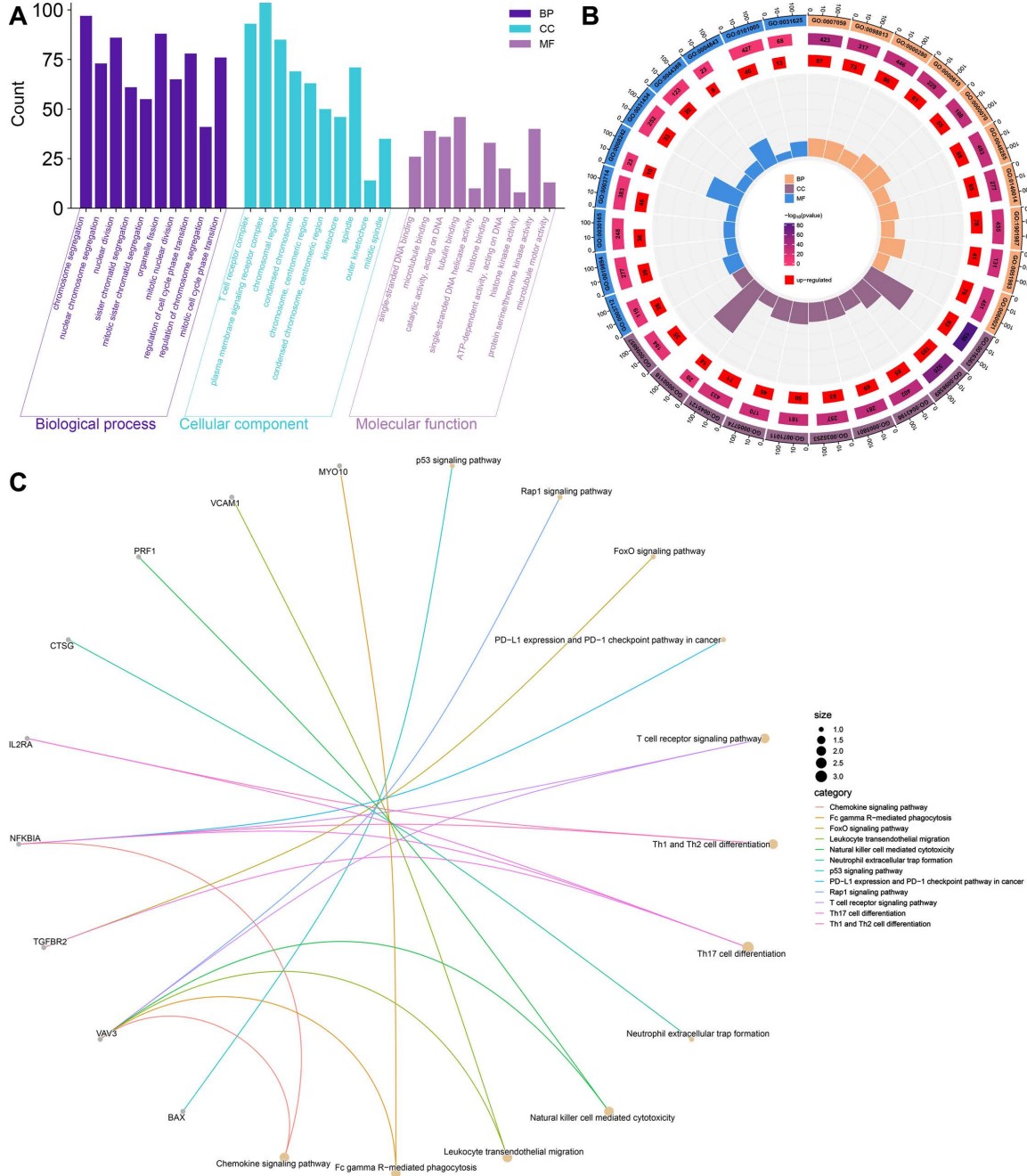

**Fig 3. Significant GO and KEGG pathway analysis of *SATB1-AS1* co-expressed genes.** (A B) GO enrichment analyses of the top 10 terms in biological processes, cellular components and molecular functions; (C) KEGG pathway analyses of the top 12 terms in species enrichment.

### 3.5 Pan-cancer analysis of *SATB1-AS1*

To assess the tissue specificity of *SATB1-AS1*, we calculated the difference in expression between normal and tumor samples in each tumor via R software and analyzed the significance of the difference via unpaired Wilcoxon rank sum and signed rank tests (Fig 6A). We observed significant upregulation in seven tumors, such as lower grade glioma and glioblastoma (GBMLGG), Brain Lower Grade Glioma(LGG), Acute Myeloid Leukemia (LAML), Pheochromocytoma & Paraganglioma (PCPG) and Kidney Chromophobe (KICH), and we observed significant downregulation in 25 tumors, such as Glioblastoma (GBM), Kidney Clear Cell Carcinoma (KIRC), Lung Squamous Cell Carcinoma (LUSC), Liver Cancer (LIHC), Melanoma (SKCM), Bladder Cancer (BLCA), Thyroid Cancer (THCA), Ovarian Cancer (OV), Pancreatic Cancer (PAAD), Testicular Cancer (TGCT), and Uterine Carcinosarcoma (UCS). Survival analysis of *SATB1-AS1* in cancers other than THYM revealed that *SATB1-AS1* also had a significant effect on survival in BLCA, KIRC, Acute Myeloid Leukemia (LAML), Lower Grade Glioma (LGG), Lung Adenocarcinoma (LUAD), Sarcoma (SARC), SKCM and Ocular melanomas (UVM) (*P*<0.05, Fig 6B).

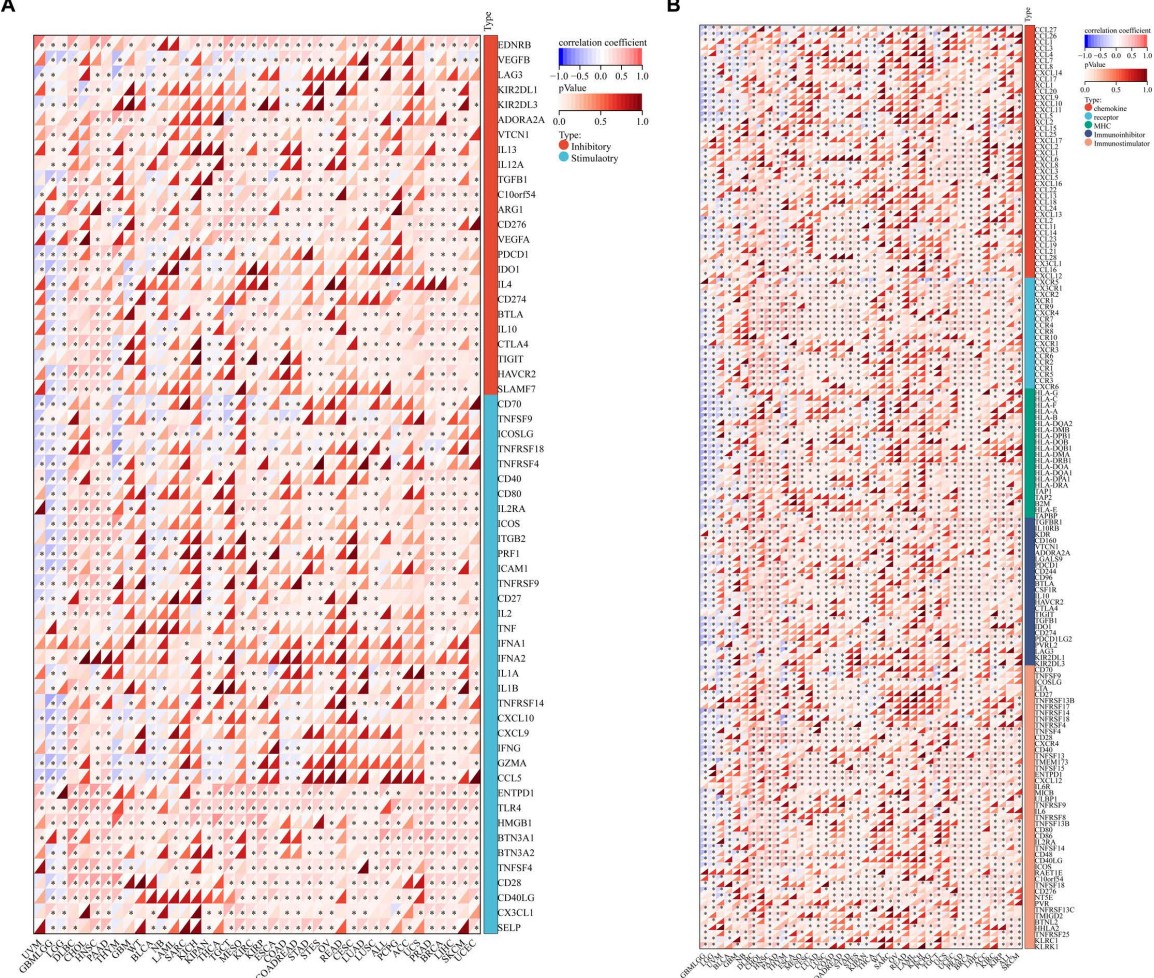

**Fig 4. *SATB1-AS1* co-expressed genes and immune genes between several cancers in spearman correlation.** (A) SATB1 gene was significantly correlated with 60 two-class immune checkpoint pathway genes (Inhibitory (24), Stimulatory (36)). (B) SATB1 gene was significantly correlated with 150 five-class immune pathways (chemokine (41), receptor (18), MHC (21), Immunoinhibitor (24) and Immunostimulator (46)) marker genes.

# 4 Discussion

Thymoma is a relatively inert tumor, but its pathogenesis is still unclear, and more features and targets are needed for early diagnosis, precise therapeutic targets, and prognosis prediction. The individualized treatment plan for THYM should be formulated by a multidisciplinary team of thoracic surgeons, medical oncologists, imaging physicians, pathologists, and

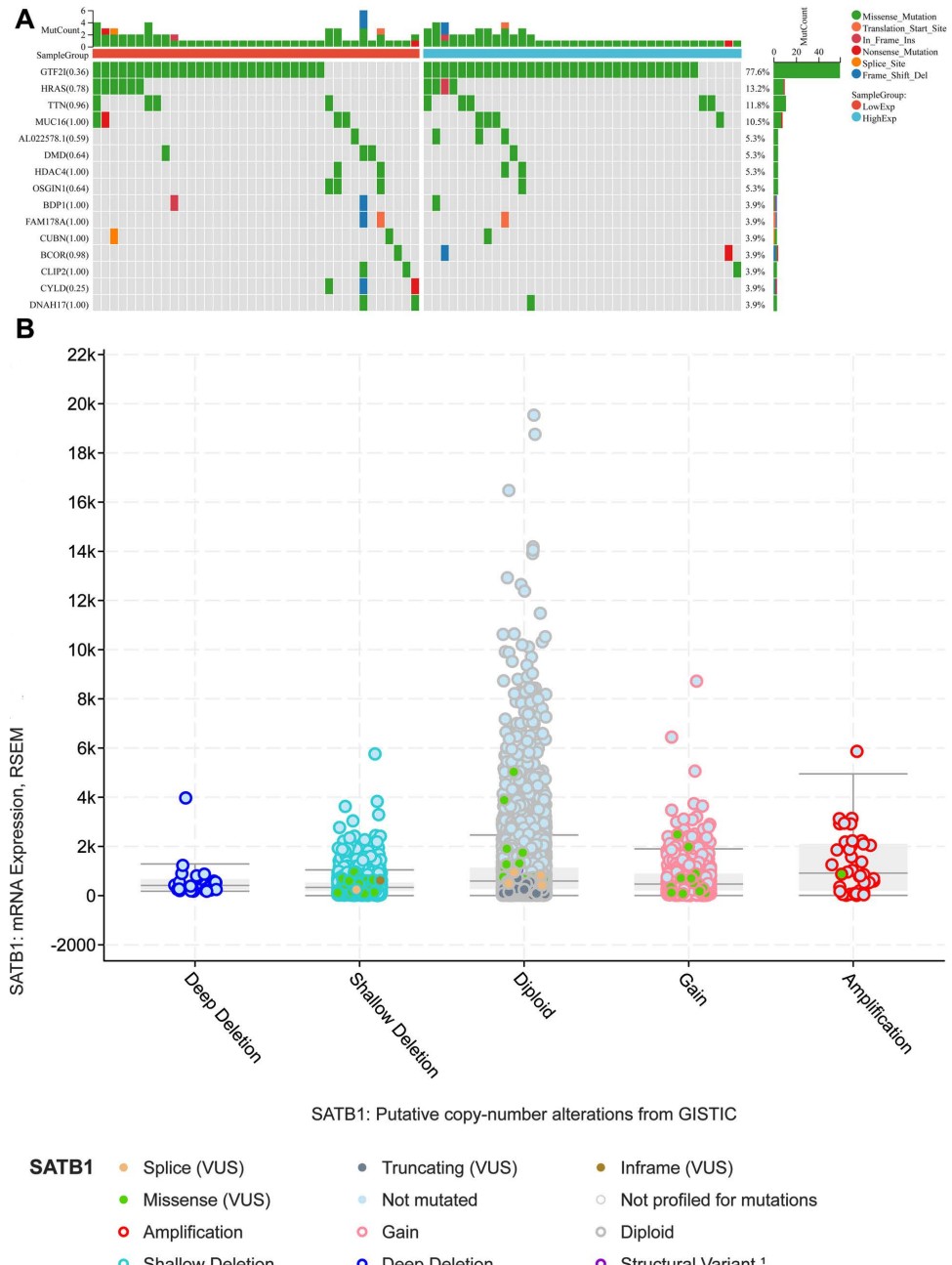

**Fig 5. Mutational landscape of _SATB1-AS1_ in thymic carcinoma.** (A) _SATB1-AS1_ co-expressed genes in THYM Tumor mutation burden (TMB) per sample. (B) The study of the origin of SATB1 in many TCGA cancers by the cBioPortal database.

radiotherapists on the basis of the patient's TNM stage and Masaoka-Koga stage to provide optimal strategies for prolonging the patient's long-term survival and reducing the side effects of treatment [3]. However, cancer treatment is limited by intra- and intertumor heterogeneity, which requires the study of mechanisms that work across patient characteristics to develop innovative therapies.

In this study, we performed bioinformatics analysis in conjunction with data collected from publicly available databases with the aim of exploring eRNAs as prognostically relevant biomarkers in THYM. On the basis of the TCGA database, 134 eRNAs with prognostic value in THYM were identified. Among them, both *SATB1-AS1* and the target gene SATB1 were strongly associated with patient prognosis. Low *SATB1-AS1* and SATB1 expression was associated with poor clinical characteristics (stage staging). However, the role of *SATB1-AS1* in prognosis was inconsistent across cancers.

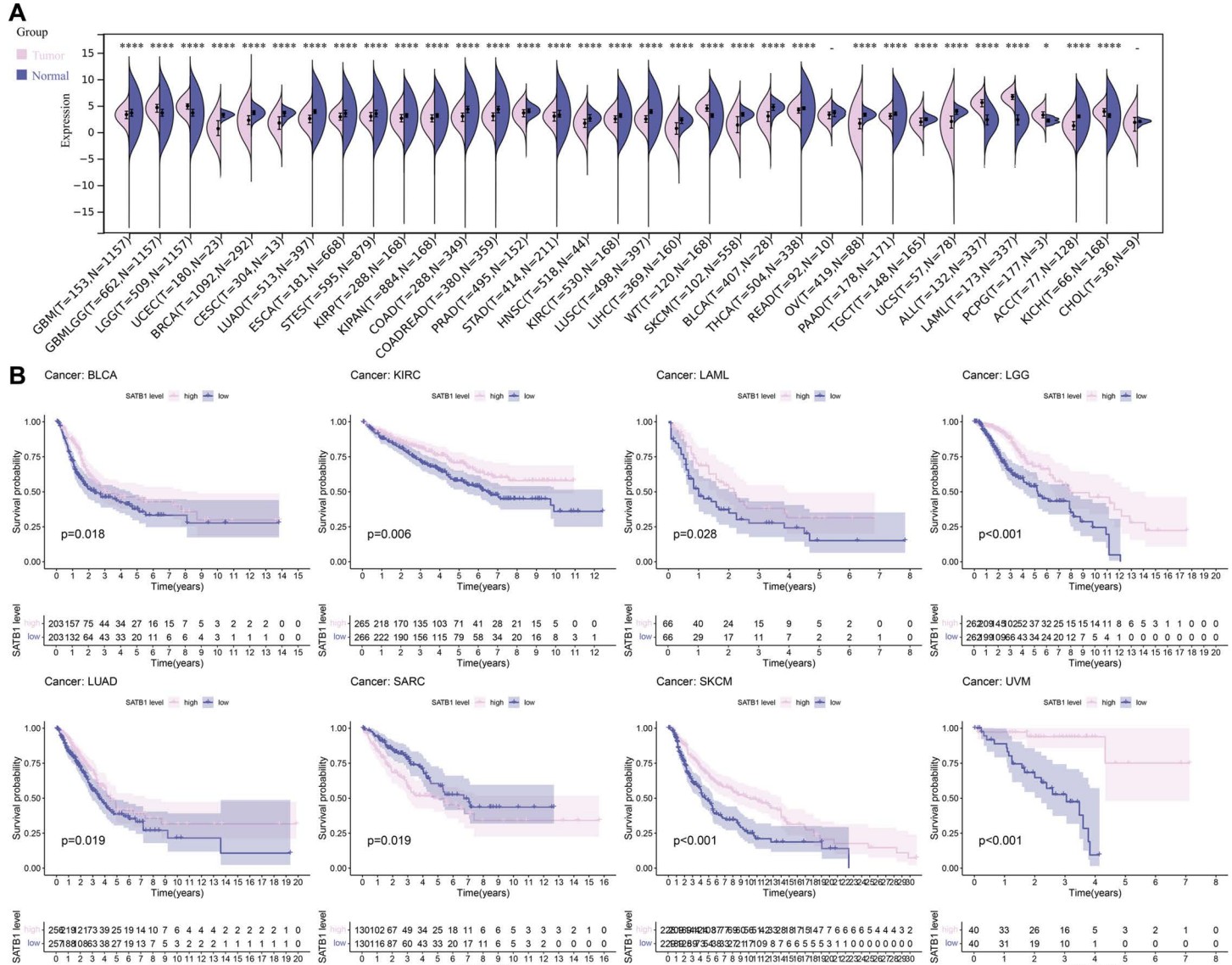

**Fig 6. Pan-cancer analysis of *SATB1-AS1*.** (A) Exploring the expression of *SATB1-AS1* in pan-cancer. (B) The prognostic effect of *SATB1-AS1* in other TCGA cancer types was investigated using UCSC Xena browser.

Low *SATB1-AS1* expression in BLCA, KIRC, LAML, LGG, LUAD, SARC, SKCM and UVM was associated with poor patient prognosis, suggesting that *SATB1-AS1* may play a role as an oncogene in tumor progression. To investigate the role of *SATB1-AS1* in the biological behavior of tumors in detail, GO enrichment analysis and KEGG pathway analysis of *SATB1-AS1* were performed in this study. The results revealed that high expression of *SATB1-AS1* was associated with the participation and regulation of leukocyte transendothelial migration, natural killer cell-mediated cytotoxicity, neutrophil extracellular trap formation, PD-L1 expression and the PD-1 checkpoint pathway in cancer, the T-cell receptor signaling pathway, Th1 and Th2 cell differentiation, Th17 cell differentiation and Th17 cell differentiation, and other molecules and pathways involved in cell biological behavior. It has been reported that tight regulation of the expression level of SATB1 isoforms and posttranslational modifications of the protein are essential for the regulatory role of SATB1 in T-cell development and that SATB1-deficient thymocytes are partially redirected to inappropriate T lineages and are unable to give rise to the NKT and Treg subpopulations [16,17]. SATB1 is a key regulator of thymocyte development. Moreover, SATB1 is a genome organizer that is highly expressed in double-positive thymocytes, and SATB1 deletion leads to a variety of defects in T-cell development, including impaired positive and negative selection and impaired differentiation of thymic regulatory T cells (Tregs) [18,19].

The eRNAs act not only as tumor suppressors by regulating the expression of tumor suppressor genes but also as major regulators of oncogenes, exhibiting potential oncogenic functions [20]. The potential role of eRNAs in mediating cancer-related enhancer functions and gene transcription is also key to understanding how the rewiring of gene expression in cancer cells occurs [21]. eRNAs are also involved in immune checkpoint-associated pathways that regulate the immune response and influence tumor development [22]. In this study, we found that *SATB1-AS1* was coexpressed with immune checkpoint-associated genes (e.g., CD-8, CCL5, and IFNG) in THYM, and in pancancer analyses, *SATB1-AS1* was also coexpressed with immune checkpoint-associated genes enriched in pathways negatively regulating immune cell proliferation and activation in a variety of cancers. These analyses suggest that *SATB1-AS1* may be a potential marker for the early diagnosis and prognosis of THYM and may be a promising immune-related therapeutic target for precision treatment of THYM patients. Although these studies provide new directions for exploring the mechanistic functions of eRNAs, many challenges remain: experiments are needed to further understand the functions and potential mechanisms of eRNAs in gene regulation and chromosomal interactions; the bioactivity of eRNAs and their correlation with diseases have not yet been fully elucidated; and, more importantly, the abundance of eRNAs in vivo is low and unstable, and more sensitive methods need to be developed to recognize eRNAs. More importantly, the abundance of eRNAs in vivo is low and unstable, and more sensitive methods need to be developed to recognize eRNAs.

Recent research shows that *SATB1-AS1* may be a potential therapeutic target for shear stress regulation [23]. The aryl hydrocarbon receptor (AhR)activation regulates the expression of lncRNA (*SATB1-AS1*) in response to the carcinogen benzo[a]pyrene [24]. In the study of acute myeloid leukemia, when the expression of long non-coding RNA SATB1-AS1 was inhibited, the proliferation ability of drug-resistant cell lines HL60/Adr and OCI-AML5/Cyt was significantly inhibited, and the apoptosis level was significantly increased [25]. A GWAS revealed that many disease-related genetic variants are enriched in enhancer elements, demonstrating the importance of eRNAs in the human genome from multiple perspectives [26]. The importance of eRNAs in the human genome has been demonstrated in many ways. Recent studies have identified wortmannin and valproic acid as potential anticancer drugs that can block eRNA activity, mainly by targeting major components of the extracellular matrix, including SPP1, COL1A1 and FN1 [27]. Thus, eRNA blockade is a valuable area for further research in cancer therapy.

## 5 Conclusion

In this study, we explored the prognosis-related genes of THYM patients as well as the related target genes, discussing the possibility of eRNAs as potential biomarkers and disease therapeutic targets as well as some directions for future research. The results revealed the following: (1) *SATB1-AS1* was identified for the first time as a highly positively

correlated eRNA with its target gene SATB 1 among the prognosis-related eRNAs in THYM. (2) *SATB1-AS1* may be an independent biomarker for the prognosis of THYM patients. (3) There was a significant correlation between SATB1-AS and immunomodulatory genes. (4) *SATB1-AS1* is significantly differentially expressed in pan-cancer assays and can be used as a prognostic marker for multiple cancers. Our findings may enhance the understanding of THYM tumorigenesis mechanisms, leading to better selection of therapeutic agents that individually target specific eRNAs and their cofactors.

## 6 Limitations

This study has several limitations. First, our study analyzed only the publicly available TCGA dataset and lacked a validation cohort to confirm the association between eRNA *SATB1-AS1* and THYM prognosis. Second, although the relevance of the eRNA *SATB1-AS1* having an immune checkpoint-associated gene was validated in pancancer, basic experiments are needed to confirm this relationship, which will be the focus of our follow-up study.

## Supporting information

**S1 Table. 133 eRNAs significantly associated with OS were identified via the Kaplan–Meier log-rank test.** (XLSX)

## Acknowledgments

Thank you to all patients and staff of TCGA and UCSC Xena database.

## Author contributions

**Conceptualization:** Hongpeng Wang.

**Data curation:** Hongpeng Wang.

**Formal analysis:** Hongpeng Wang.

**Funding acquisition:** Lei Jiang.

**Investigation:** Yichu Huang.

**Methodology:** Yichu Huang.

**Project administration:** Yichu Huang.

**Software:** Guangtao Min.

**Supervision:** Guangtao Min.

**Validation:** Guangtao Min.

**Writing – original draft:** Hongpeng Wang.

**Writing – review & editing:** Lei Jiang.

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
