## [Decision Letter · Decision Letter 0]

12 Mar 2025

PONE-D-25-07416Identification of the immune-associated enhancer RNA SATB1-AS1 as a novel biomarker for thymic cancer prognosisPLOS ONE

Dear Dr. jiang,

Thank you for submitting your manuscript to PLOS ONE. After careful consideration, we feel that it has merit but does not fully meet PLOS ONE’s publication criteria as it currently stands. Therefore, we invite you to submit a revised version of the manuscript that addresses the points raised during the review process.

We look forward to receiving your revised manuscript.

Kind regards,

Benjamin Benzon, Ph.D., M.D.

Academic Editor

PLOS ONE

https://ddrare.nibiohn.go.jp/cgi-bin/clinical_trials.cgi?db=drug_who&disease_id=96

In your revision ensure you cite all your sources (including your own works), and quote or rephrase any duplicated text outside the methods section. Further consideration is dependent on these concerns being addressed.

5. Please provide a complete Data Availability Statement in the submission form, ensuring you include all necessary access information or a reason for why you are unable to make your data freely accessible. If your research concerns only data provided within your submission, please write "All data are in the manuscript and/or supporting information files" as your Data Availability Statement.

6. PLOS requires an ORCID iD for the corresponding author in Editorial Manager on papers submitted after December 6th, 2016. Please ensure that you have an ORCID iD and that it is validated in Editorial Manager. To do this, go to ‘Update my Information’ (in the upper left-hand corner of the main menu), and click on the Fetch/Validate link next to the ORCID field. This will take you to the ORCID site and allow you to create a new iD or authenticate a pre-existing iD in Editorial Manager.

Additional Editor Comments:

Very nice study, reviewers have made specific comments please address them. I am looking forward to your reply and revised version of the manuscript.

Reviewers' comments:

Reviewer's Responses to Questions

**Comments to the Author**

1. Is the manuscript technically sound, and do the data support the conclusions?

Reviewer #1: Partly

Reviewer #2: Partly

Reviewer #3: Yes

2. Has the statistical analysis been performed appropriately and rigorously? 

Reviewer #1: Yes

Reviewer #2: Yes

Reviewer #3: Yes

3. Have the authors made all data underlying the findings in their manuscript fully available?

Reviewer #1: Yes

Reviewer #2: Yes

Reviewer #3: Yes

4. Is the manuscript presented in an intelligible fashion and written in standard English?

Reviewer #1: No

Reviewer #2: No

Reviewer #3: Yes

5. Review Comments to the Author

***Reviewer #1:*** The study analyzes enhancer RNAs in thymoma using data from the TCGA database and explores their potential prognostic and therapeutic roles.

Comments:

1. The identification of a novel thymoma biomarker could be highly valuable. However, a major weakness of this study is that the authors rely solely on database analysis without any supporting in vitro or in vivo experiments. The manuscript only establishes a correlation between eRNA SATB1-AS1 and thymoma disease stage. Therefore, the claim in the title that SATB1-AS1 is "a novel biomarker for thymic cancer prognosis" appears to be an overstatement.

2. The manuscript lacks fluency and contains some errors. For example, in section 2.2, the first sentence is repeated twice. Careful proofreading and editing are needed to improve readability.

3. The placement of figures could be improved. I recommend positioning them after the relevant result descriptions rather than grouping them between the results and discussion sections. Additionally, the figures have low resolution, making it difficult to read the text, particularly in Figure 1.

Conclusion:

While the discovery of a new thymoma biomarker could have significant clinical implications, additional in vitro, in vivo, or epidemiological studies are necessary to validate its prognostic potential.

***Reviewer #2:*** In their manuscript titled "Identification of the immune-associated enhancer RNA SATB1-AS1 as a novel

biomarker for thymic cancer prognosis" Wang et al. have performed bioinformatic and statistical analyses on publicly available TCGA data to determine the roles of enhancer RNAs, specifically SATB1-AS1, in thymomas and other tumors. I have several comments that must be addressed prior to the publication of this manuscript.

GENERAL

The symbols for genes and RNAs should be written in the italic font style (proteins should be written in regular style).

ABSTRACT

Methods: “Validation of the prognostic potential of target eRNAs across cancers in other cancer types.” This sentence has no verb; perhaps “was performed” is missing at the end?

Results: “SATB1-AS1 and the target gene SATB1 were associated with poor patient prognosis” This statement is misleading as high SATB1-AS1 expression wasa positive prognostic marker!

“…SATB1-AS1 expression was correlated with patient stage…” You should add “negatively” in front of correlated.

INTRODUCTION

-“Surgery is the treatment of choice for tumor management. Surgery, as the treatment of choice for tumor therapy…” No need to write the same information twice.

-“It is also involved in driving the transcription of target genes in various ways, such as three-dimensional chromatin remodeling and enhancer‒promoter loops.” Again, this was already stated in the previous sentence and should be removed.

-lncRNA, miRNA, SCCHN, and ESCA should be written in full form the first time they are mentioned.

-”A recent study also revealed that eRNA transcripts are not associated with oncogenic genes.” No reference is provided.

MATERIALS AND METHODS

2.1 The exact numbers of tumor patients and controls for which expression data was obtained should be stated.

2.2 “The R package "limma" was used to screen for differentially expressed eRNAs…” This is stated twice.

2.5 A closing bracket is missing after “stimulatory (36)”

2.7 Which test was used to test the normality of data distribution? This information should be added.

RESULTS

3.2 SATB1-AS1 or SATB1-as1? You need to be consistent with terminology.

3.2 “higher SATB1-AS1 expression was more strongly associated with tumor grade and lymphatic infiltration than lower SATB1-AS1 expression was, and lower SATB1-AS1 expression was more strongly associated with tumor grade and lymphatic infiltration.” This statement contradicts itself, so it is not clear what the authors meant to convey.

3.2 “We found that high SATB1-AS1 expression was associated with a favorable prognosis in OV patients.” Why is this mentioned in this section?

3.2 “we also found that SATB1-AS1 expression levels were independent of age, sex, and tumor grade.” Figure 2B clearly demonstrates that expression levels are significantly lower in stage III and IV tumors compared to stage I and II. Also, the text mentions tumor grade but the graph is annotated as tumor stage. These terms are not identical, so you should correct whichever one is wrong.

3.3 The categories of GO enrichment should be written in full form since they are mentioned the first time here.

3.5 Again, full forms of the tumors should be written since they are mentioned the first time here.

FIGURES

Figure 1: The resolution is too small and I cannot tell what is written on the axes or the R and p values on the graphs. The figure should be larger and of a higher resolution.

Figure 2B-E: The description states SATB1-AS1, but SATB1 is written on the y-axis of the graphs. This should be corrected.

Figure 3: The text should be larger, especially on 3B, and everything should be of a higher resolution.

Figure 4: The same as Figures 1 and 3, the text is too small, low resolution, everything is smushed together. In my opinion, this Figure, even when corrected, does not provide any valuable information and the data contained inside would be better suited for a supplementary table.

Figures 5 and 6: Same comment as Figures 1 and 3.

DISCUSSION

-“The optimal strategy for prolonging long-term survival and minimizing the side effects of treatment should be developed by a multidisciplinary team of thoracic surgeons” No need to write the same information twice.

-“However, cancer treatment is limited by both intratumor and intertumor factors. However,

cancer treatment is limited by intra- and intertumor heterogeneity” Same as above.

***Reviewer #3:*** Review of manuscript by Wang et al. “Identification of the immune-associated enhancer RNA SATB1-AS1 as a novel biomarker for thymic cancer prognosis”

In the study by Wang et al., the authors performed a bioinformatics analysis using publicly available databases to explore the role of enhancer RNAs (eRNAs) as prognostic biomarkers in thymoma. The study successfully identified the immune-associated enhancer RNA SATB1-AS1 as a novel biomarker for thymic cancer prognosis.

This manuscript is well-written and easily understood. The introduction provides a clear and concise background, giving sufficient context for the study. The materials and methods section is well-structured and described in adequate detail. The manuscript can be accepted for publication after addressing the following comments.

Specific Comments:

1. Data Extraction:

o Please specify the date when the data was extracted from the TCGA databases and Xena.

2. Figure Quality and Assembly:

o Please provide details on how the figures were assembled.

o The quality of the figures is low, making it difficult to distinguish details. Please improve the resolution of Figures 1, 3, and 4. If the issue is due to the PDF format of the manuscript, this comment can be disregarded.

3. Discussion:

o There appears to be redundancy in the following sentence; please revise it for clarity: "However, cancer treatment is limited by both intratumor and intertumor factors. However, cancer treatment is limited by intra- and intertumor heterogeneity, which requires the study of mechanisms that work across patient characteristics to develop innovative therapies."

o Please include references and elaborate on the known information about SATB1-AS1 and its connection to other known tumors.

4. Conclusion:

o The following sentence appears incomplete; please revise for clarity and coherence: (4) Differential expression of SATB1-AS1 across cancers and its impact on survival.

After addressing these comments, the manuscript will be suitable for publication.

6. PLOS authors have the option to publish the peer review history of their article (what does this mean? ). If published, this will include your full peer review and any attached files.

**Do you want your identity to be public for this peer review?** For information about this choice, including consent withdrawal, please see our Privacy Policy .

Reviewer #1: No

Reviewer #2: No

Reviewer #3: No

---

## [Author Response · Author response to Decision Letter 1]

27 Mar 2025

Dear Editors and Reviewers:

Thank you for your letter and for the reviewers’ comments concerning our manuscript entitled “Identification of the immune-associated enhancer RNA SATB1-AS1 as a novel biomarker for thymic cancer prognosis” (ID: PONE-D-25-07416). Those comments are all valuable and very helpful for revising and improving our paper, as well as the important guiding significance to our researches. We have studied comments carefully and have made correction which we hope meet with approval. Revised portion are marked with different colors in the paper. The main corrections in the paper and the responds to the reviewer’s comments are as flowing:

Reviewer #1: The study analyzes enhancer RNAs in thymoma using data from the TCGA database and explores their potential prognostic and therapeutic roles.

Comments:

1.The identification of a novel thymoma biomarker could be highly valuable. However, a major weakness of this study is that the authors rely solely on database analysis without any supporting in vitro or in vivo experiments. The manuscript only establishes a correlation between eRNA SATB1-AS1 and thymoma disease stage. Therefore, the claim in the title that SATB1-AS1 is "a novel biomarker for thymic cancer prognosis" appears to be an overstatement.

Response Thanks for your suggestion. Due to the limitations of the college and time constraints, this study is temporarily unable to provide more experiments to verify the expression regulation of key genes in thymic cancer. In the future, we will conduct in-depth experiments to verify the specific regulatory mechanisms between SATB1-AS1 and thymus cancer. As shown in Figure 6, SATB1-AS1 showed significantly different expression in various malignant tumors. In particular, the expression level of this molecule was significantly lower in thymus cancer tissues than in tumor control tissues (P <0.0001). Survival analysis revealed that thymus cancer patients with high expression of SATB1-AS1 showed a better clinical prognosis, and their survival curves were statistically significant different from those in the low expression group (Figure 1). Further staging studies showed that patients with low expression of SATB1-AS1 had generally higher tumor stages, suggesting that this molecule may be involved in biological processes regulating tumor progression. This study speculated that long non-coding RNA SATB1-AS1 might play a role in the development of thymus carcinoma by regulating the expression of SATB1 gene.

2. The manuscript lacks fluency and contains some errors. For example, in section 2.2, the first sentence is repeated twice. Careful proofreading and editing are needed to improve readability.

Response Thanks for your suggestion. We have been checked to make the article more readable. And fixed 2.2 duplication problem.

3. The placement of figures could be improved. I recommend positioning them after the relevant result descriptions rather than grouping them between the results and discussion sections. Additionally, the figures have low resolution, making it difficult to read the text, particularly in Figure 1.

Response Thanks for your suggestion. We changed the position of the image and improved the sharpness of the corresponding image

Conclusion:

While the discovery of a new thymoma biomarker could have significant clinical implications, additional in vitro, in vivo, or epidemiological studies are necessary to validate its prognostic potential.

Response Thanks for your suggestion. Due to the limitations of the college and time constraints, this study is temporarily unable to provide more experiments to verify the expression regulation of key genes in thymic cancer. In the future, we will conduct in-depth experiments to verify the specific regulatory mechanisms between SATB1-AS1 and thymus cancer.

Reviewer #2: In their manuscript titled "Identification of the immune-associated enhancer RNA SATB1-AS1 as a novel biomarker for thymic cancer prognosis" Wang et al. have performed bioinformatic and statistical analyses on publicly available TCGA data to determine the roles of enhancer RNAs, specifically SATB1-AS1, in thymomas and other tumors. I have several comments that must be addressed prior to the publication of this manuscript.

GENERAL

The symbols for genes and RNAs should be written in the italic font style (proteins should be written in regular style).

Response Thanks for your suggestion. We did careful proofreading of genes 、 proteins and RNA writing

ABSTRACT

Methods: “Validation of the prognostic potential of target eRNAs across cancers in other cancer types.” This sentence has no verb; perhaps “was performed” is missing at the end?

Response Thanks for your suggestion. We changed to “Finally, the prognostic potential of the eRNA was validated in other cancer types.”

Results: “SATB1-AS1 and the target gene SATB1 were associated with poor patient prognosis” This statement is misleading as high SATB1-AS1 expression was a positive prognostic marker!

Response Thanks for your suggestion. We changed to “SATB1-AS1 and the target gene SATB1 high expression were associated with a positive patient prognosis”

“…SATB1-AS1 expression was correlated with patient stage…” You should add “negatively” in front of correlated.

Response Thanks for your suggestion. We changed to “SATB1-AS1 expression was negatively correlated with patient stage”

INTRODUCTION

-“Surgery is the treatment of choice for tumor management. Surgery, as the treatment of choice for tumor therapy…” No need to write the same information twice.

Response Thanks for your suggestion. We delated “Surgery is the treatment of choice for tumor management.”

-“It is also involved in driving the transcription of target genes in various ways, such as three-dimensional chromatin remodeling and enhancer‒promoter loops.” Again, this was already stated in the previous sentence and should be removed.

Response Thanks for your suggestion. We delated “It is also involved in driving the transcription of target genes in various ways, such as three-dimensional chromatin remodeling and enhancer‒promoter loops.”

-lncRNA, miRNA, SCCHN, and ESCA should be written in full form the first time they are mentioned.

Response Thanks for your suggestion. We added the full lncRNA and miRNA name and deleted SCCHN, and ESCA.

-”A recent study also revealed that eRNA transcripts are not associated with oncogenic genes.” No reference is provided.

Response Thanks for your suggestion. We delated “A recent study also revealed that eRNA transcripts are not associated with oncogenic genes.”

MATERIALS AND METHODS

2.1 The exact numbers of tumor patients and controls for which expression data was obtained should be stated.

Response Thanks for your suggestion. We changed to “We first screened 124 THYM-related patients and 14 normal sample via the TCGA website and downloaded their transcript data.”

2.2 “The R package "limma" was used to screen for differentially expressed eRNAs…” This is stated twice.

Response Thanks for your suggestion. We delated “The R package "limma" was used to screen for differentially expressed eRNAs”

2.5 A closing bracket is missing after “stimulatory (36)”

Response Thanks for your suggestion. We added a closing bracket.

2.7 Which test was used to test the normality of data distribution? This information should be added.

Response Thanks for your suggestion. We added “The Shapiro-Wilk test is used to test the normality of the data distribution”

RESULTS

3.2 SATB1-AS1 or SATB1-as1? You need to be consistent with terminology.

Response Thanks for your suggestion. We unify the description of SATB1-AS1

3.2 “higher SATB1-AS1 expression was more strongly associated with tumor grade and lymphatic infiltration than lower SATB1-AS1 expression was, and lower SATB1-AS1 expression was more strongly associated with tumor grade and lymphatic infiltration.” This statement contradicts itself, so it is not clear what the authors meant to convey.

Response Thanks for your suggestion. We changed to “Furthermore, lower expression of SATB1-AS1 is more associated with tumor stage and lymphatic invasion compared to higher expression (Fig 2B). Notably, the expression level of SATB1-AS1 is unrelated to age, gender, and cancer status (Fig 2C-E)”

3.2 “We found that high SATB1-AS1 expression was associated with a favorable prognosis in OV patients.” Why is this mentioned in this section?

Response Thanks for your suggestion. We changed to “We found that high SATB1-AS1 expression was associated with a favorable prognosis in THYM patients”

3.2 “we also found that SATB1-AS1 expression levels were independent of age, sex, and tumor grade.” Figure 2B clearly demonstrates that expression levels are significantly lower in stage III and IV tumors compared to stage I and II. Also, the text mentions tumor grade but the graph is annotated as tumor stage. These terms are not identical, so you should correct whichever one is wrong.

Response Thanks for your suggestion. We corrected it to “Furthermore, lower expression of SATB1-AS1 is more associated with tumor stage and lymphatic invasion compared to higher expression (Fig 2B). Notably, the expression level of SATB1-AS1 is unrelated to age, gender, and cancer status (Fig 2C-E)”

3.3 The categories of GO enrichment should be written in full form since they are mentioned the first time here.

Response Thanks for your suggestion. GO and KEGG enrichment analysis have been mentioned in the methods section and their full names have been written.

3.5 Again, full forms of the tumors should be written since they are mentioned the first time here.

Response Thanks for your suggestion. We added all full forms of the tumors.

FIGURES

Figure 1: The resolution is too small and I cannot tell what is written on the axes or the R and p values on the graphs. The figure should be larger and of a higher resolution.

Response Thanks for your suggestion. We have updated the resolution of all the pictures and submitted the original pictures in the attachment

Figure 2B-E: The description states SATB1-AS1, but SATB1 is written on the y-axis of the graphs. This should be corrected.

Response Thanks for your suggestion. We corrected it to “SATB1-AS1”

Figure 3: The text should be larger, especially on 3B, and everything should be of a higher resolution.

Response Thanks for your suggestion. We have updated the resolution of all the pictures and submitted the original pictures in the attachment

Figure 4: The same as Figures 1 and 3, the text is too small, low resolution, everything is smushed together. In my opinion, this Figure, even when corrected, does not provide any valuable information and the data contained inside would be better suited for a supplementary table.

Response Thanks for your suggestion. Functional enrichment found that SATB1 was associated with immunity, and Figure 4 mainly showed its correlation with immune checkpoints. So we don't think he should be used as a supplement.

Figures 5 and 6: Same comment as Figures 1 and 3.

Response Thanks for your suggestion. We have updated the resolution of all the pictures and submitted the original pictures in the attachment

DISCUSSION

-“The optimal strategy for prolonging long-term survival and minimizing the side effects of treatment should be developed by a multidisciplinary team of thoracic surgeons” No need to write the same information twice.

Response Thanks for your suggestion. We removed the duplicate content.

-“However, cancer treatment is limited by both intratumor and intertumor factors. However, cancer treatment is limited by intra- and intertumor heterogeneity” Same as above.

Response Thanks for your suggestion. We removed the duplicate content.

Reviewer #3: Review of manuscript by Wang et al. “Identification of the immune-associated enhancer RNA SATB1-AS1 as a novel biomarker for thymic cancer prognosis”

In the study by Wang et al., the authors performed a bioinformatics analysis using publicly available databases to explore the role of enhancer RNAs (eRNAs) as prognostic biomarkers in thymoma. The study successfully identified the immune-associated enhancer RNA SATB1-AS1 as a novel biomarker for thymic cancer prognosis.

This manuscript is well-written and easily understood. The introduction provides a clear and concise background, giving sufficient context for the study. The materials and methods section is well-structured and described in adequate detail. The manuscript can be accepted for publication after addressing the following comments.

Specific Comments:

1. Data Extraction:

o Please specify the date when the data was extracted from the TCGA databases and Xena.

Response Thanks for your suggestion. We added “The data extraction deadline is August 24, 2023”

2. Figure Quality and Assembly:

o Please provide details on how the figures were assembled.

Response Thanks for your suggestion. We draw the figures based on R software and the database below.

o The quality of the figures is low, making it difficult to distinguish details. Please improve the resolution of Figures 1, 3, and 4. If the issue is due to the PDF format of the manuscript, this comment can be disregarded.

Response Thanks for your suggestion. We have updated the resolution of all the pictures and submitted the original pictures in the attachment

3. Discussion:

o There appears to be redundancy in the following sentence; please revise it for clarity: "However, cancer treatment is limited by both intratumor and intertumor factors. However, cancer treatment is limited by intra- and intertumor heterogeneity, which requires the study of mechanisms that work across patient characteristics to develop innovative therapies."

Response Thanks for your suggestion. We removed the duplicate content.

o Please include references and elaborate on the known information about SATB1-AS1 and its connection to other known tumors.

Response Thanks for your suggestion. We added “Recent research shows that SATB1-AS1 may be a potential therapeutic target for shear stress regulation. The aryl hydrocarbon receptor (AhR) activation regulates the expression of lncRNA (SATB1-AS1) in response to the carcinogen benzo[a]pyrene. In the study of acute myeloid leukemia, when the expression of long non-coding RNA SATB1-AS1 was inhibited, the proliferation ability of drug-resistant cell lines HL60/Adr and OCI-AML5/Cyt was significantly inhibited, and the apoptosis level was significantly increased.”

4. Conclusion:

o The following sentence appears incomplete; please revise for clarity and coherence: (4) Differential expression of SATB1-AS1 across cancers and its impact on survival.

Response Thanks for your suggestion. We changed to “SATB1-AS1 is significantly differentially expressed in pan-cancer assays and can be used as a prognostic marker for multiple cancers.”

After addressing these comments, the manuscript will be suitable for publication.

Special thanks to you for your good comments.

We tried our best to improve the manuscript and made some changes in the manuscript. These changes will not influence the content and framework of the paper.

And here we did not list the changes but marked marked with different colors in revised paper.

We appreciate for Editors/Reviewers warm work earnestly, and hope that the correction will meet with approval. Once again, thank you very much for your comments and suggestions.

---

## [Editor Report · Decision Letter 1]

2 Apr 2025

Identification of the immune-associated enhancer RNA SATB1-AS1 as a novel biomarker for thymic cancer prognosis

PONE-D-25-07416R1

Dear Dr. jiang,

We’re pleased to inform you that your manuscript has been judged scientifically suitable for publication and will be formally accepted for publication once it meets all outstanding technical requirements.

Kind regards,

Benjamin Benzon, Ph.D., M.D.

Academic Editor

PLOS ONE
---

## [Editor Report · Acceptance letter]

PONE-D-25-07416R1

PLOS ONE

Dear Dr. Jiang,

I'm pleased to inform you that your manuscript has been deemed suitable for publication in PLOS ONE. Congratulations! Your manuscript is now being handed over to our production team.

Kind regards,

on behalf of

Dr. Benjamin Benzon

Academic Editor

PLOS ONE